# From biorepositories to data repositories: Open-access resources accelerate early R&D and validation of equitable diagnostic tools

**Roger Peck**[‡*], **Helen L. Storey**[‡*], **Becky Barney, Shirli Israeli**[¤]**, Olivia Halas, Deborah Oroszlan, Shiri Brodsky, Neha Agarwal, Eileen Murphy, Mariana Sagalovsky, Jessica Cohen, Elizabeth Trias, Aaron Schutzer, David S. Boyle**

PATH, Seattle, Washington, United States of America

¤ Current address: Lyell, Seattle, Washington, United States of America
‡ RP and HLS are co-first authors on this work.
* rpeck@path.org (RP); hstorey@path.org (HLS)

## Abstract

Diagnostics are critical tools that guide clinical decision-making for patient care and support disease surveillance. Despite its importance, developers and manufacturers often note that access to specimen panels and essential reagents is one of the key challenges in developing quality diagnostics, particularly in low-resource settings. A recent example, as the COVID-19 pandemic unfolded there was a need for clinical samples across the globe to support the rapid development of diagnostics. To address these challenges and gaps, PATH, a global nonprofit, along with its partners collaborated to create a COVID-19 biorepository to improve access to biological samples. Since then, the need for data resources to advance universal rapid diagnostic test (RDT) readers and noninvasive clinical measurement tools for screening children have also been identified and initiated. From biospecimens to data files, there are more similarities than differences in creating open-access repositories. And to ensure equitable technologies are developed, diverse sample panels and datasets are critical in the development process. Here we share one experience in creating open-access repositories as a case study to describe the steps taken, the key factors required to establish a biorepository, the ethical and legal frameworks that guided the initiative and the lessons learned. As diagnostic tools are evolving, more forms of data are critical to de-risk and accelerate early research and development (R&D) for products serving low resource settings. Creating physical and virtual repositories of freely available, well characterized, and high quality clinical and electronic data resources defray development costs to improve equitable access and test affordability.

**Data Availability Statement:** Biospecimens referred to in this manuscript are available at request through the PATH repository website:

## Introduction

According to the World Health Organization, diagnostics "are essential for advancing universal health coverage, addressing health emergencies, and promoting healthier populations" [1]. These critical tools guide clinical decision-making for patient care, reduce the use of prescription

https://www.path.org/programs/diagnostics/washington-covid-19-biorepository/.

**Funding:** This work was funded by the Bill and Melinda Gates Foundation (https://www.gatesfoundation.org/) via the following grants: INV-016821 (RP, HLS, BB, SI, OH, DO, SB, NA, EM, MS, JC, ET, AS, DB) and INV-048193 (HLS, DO, SB). The funder did not have any additional role in the study design, data collection and analysis, decision to publish, or preparation of the manuscript.

**Competing interests:** The authors have declared that no competing interests exist.

drugs, curb the spread of drug resistance, support disease surveillance programs, and raise early alerts of a potential pandemic [2, 3]. As in the COVID-19 response, accurate and timely diagnosis has been a critical cornerstone of control efforts [4], as these tools reveal disease location and its scale, contribute to contact tracing, guide vaccination rollouts, and advance global surveillance efforts. Despite their great importance, equitable access to much-needed diagnostic tools has been an ongoing challenge [5]. In some low resource settings, diagnostics may be either unavailable in the region, too expensive, or too resource-intensive for local health systems to acquire and use. For example, the beginning of the COVID-19 pandemic, the only diagnostic tools available were lab-based tests, which were complex, costly, and infrastructure intensive. Additionally, the relative lack of regional manufacturing capacity further limits the supply of affordable and appropriate products to lower and middle income (LMIC) markets [6, 7].

Developers and manufacturers often note that access to specimen panels and essential reagents is one of the key challenges in developing quality diagnostics, particularly during epidemics. This has historically delayed the development and evaluation of critical diagnostic technologies. Creating open-access biorepositories has been an essential solution to catalyzing early research on many diseases and accelerating the development of new technologies. There are disease and population specific biorepositories all over the world [8]. An additional consideration for biorepositories focused on diseases predominately occurring in global south settings is ensuring appropriate ownership and attribution to scientists and communities contributing invaluable samples to research and biobanks when they are housed and overseen in global north geographies.

Additionally, diagnostic tools are rapidly evolving to utilize more forms of clinical samples and data to better detect health conditions, as well as translate data to action faster through connected capabilities, particularly with the development of smartphone-based screening technologies [9, 10]. For example, there is a growing interest by larger diagnostic companies in developing RDT readers and applications to deploy alongside COVID-19 RDTs. When designed with an understanding of users and local context, RDT readers have the potential to digitize diagnostic data faster and more reliably, allowing facilities and health systems to better inform decision making, as well as providing training and streamlined data entry support for users [11].

PATH, a nonprofit organization committed to global health equity, enables product development for low resource settings by building partnerships between researchers in disease endemic regions and quality manufacturers interested in LMIC markets. To accomplish this goal, PATH maintains long-standing and trusted relationships with a wide range of partners including but not limited to entities in academia, national and subnational governments, nonprofit organizations, civil society organizations, global normative bodies, and industry. When COVID-19 started in Seattle, PATH along with its partners addressed the diagnostic gap by creating one of the first COVID-19 open-access biorepositories [12]. PATH leveraged existing local resources to create the biorepository with scientists and communities in the region to accelerate tools for use globally. In sharing the key steps taken, and lessons learned from the experience, the aim is to further discussions on best practices for developing, operating, and sustaining repositories for clinical samples and data. Additionally, there is a critical need and challenge with expanding to broader data sources as diagnostic tools leverage information in new formats such as images and videos.

## Materials and methods

### Ethics statement

The Office of Research Affairs (ORA) at PATH reviewed and provided ethical approval for this work (IRBNetID 1584172). Written consent was required for samples from human

subjects' research (HSR) studies. Samples were collected from 2020–2022, and the repository continues to accept requests. No identifiable information for any samples was received by PATH.

## The Washington COVID-19 biorepository

When COVID-19 was detected in the Seattle area of Washington State—the first location in the United States—PATH had a unique opportunity to accelerate the development and validation of quality in-vitro diagnostics for COVID-19 by creating a biorepository of COVID-19 clinical specimens. With PATH's laboratories located in Seattle, in-house assets were leveraged, such as its Biosafety Level (BSL) 2 laboratory, local partnerships, ethics review committee, and legal counsel, to rapidly scale the COVID-19 biorepository.

---

**Checklist to launch the biorepository\*:**

✓ Ethical oversight

✓ Collaboration requirements

✓ Governance structure (legal, ethical and scientific)

✓ Processing and handling of biospecimens

✓ Administrative and logistical operations

✓ Communication and dissemination

\**Not necessarily specific to Covid19*

---

**Ethical oversight.** One of the earliest considerations in setting up the biorepository was ensuring proper ethical oversight for the purpose of the biorepository. PATH has an in-house program that provides scientific and ethical review for all research, with an emphasis on human subject research. Early discussions identified the need for an ethical review and approval process of the activities to create the biorepository, that would include creation of a biorepository governance plan.

**Collaboration.** Existing relationships with clinical partners across Seattle were utilized to create a repository of inactivated virus and clinical samples, including nasal swabs, tongue swabs, nasopharyngeal swabs, serum, and plasma. These partners included The Everett Clinic —Part of Optum, UW School of Medicine, FidaLab, Northwest Pathology, Washington State Public Health Laboratories and Bloodworks Northwest. Existing partnerships were critical to obtaining samples when specimens were limited, as well as at no cost when specimens were too costly for smaller diagnostic developers. The details of the partnership also varied slightly across partners. Some differences include how the samples were obtained, data accompanying the samples, the population from where the samples came, and ongoing communication with the partner about biorepository activities. The collaboration was named the Washington COVID-19 biorepository to represent the statewide effort that enabled the biorepository to scale rapidly and open its doors on March 26, 2020.

**Governance.** PATH's Office of Research Affairs (ORA) had processes for Biorepository Governance Plans (BGP) to outline the scientific, ethical, and legal oversight mechanisms that govern sample collection, storage, and distribution for future research. The project team and ORA worked collaboratively to establish the BGP framework, including key tenets for accepting samples from different types of sources. These source types included, but were not limited to, samples from human subjects' research (HSR) studies, from laboratories conducting clinical testing including clinical discards [13], and from research laboratories manipulating infectious material of COVID-19. The BGP also outlined the type of research the samples could be used to support, with a focus on COVID-19 diagnostic development, assessment of diagnostic performance, and basic research into the human immune response to COVID-19. Biospecimens from the repository cannot be sold to any recipients. To transfer samples, a process of material transfer agreements (MTAs) was used. Additional mechanisms, such as a governance committee were defined in the BGP pertaining to the ethical and scientific oversight of the biorepository samples to facilitate efficient processing of sample requests. A version of the BGP is available in the supplemental material (S1 Text).

**Biospecimens.** The PATH BSL2 laboratory in Seattle had capacity to add the biorepository, utilizing about 10% of its freezer space to host COVID-19 specimens compared to the other collections of specimens for product development research. Samples were collected with a confirmed diagnosis of COVID-19 in two distinct cohorts. The first cohort was specimens to support the development of virus-based diagnostics (e.g., antigen or nucleic acid tests) that were collected from routine clinical care at clinics throughout the region. These clinical samples were from patients with a confirmed diagnosis of COVID-19 using a Food and Drug Administration (FDA) Emergency Use Authorization (EUA) COVID-19 reverse transcription-polymerase chain reaction (RT PCR) diagnostic test and shared by the following partners: the Everett Clinic- Part of Optum, FidaLab, Northwest Pathology, Bloodworks Northwest, Washington State Public Health Laboratories and the UW School of Medicine. The nasal specimens, which made up most of the biorepository, were frozen by partnering labs then sent to PATH for cataloging and further freezer storage. All samples were deidentified prior to delivery to PATH's labs to ensure patient privacy. Additionally, freeze/thaw cycles of any specimens were tracked through the specimen ID.

The second cohort of samples (serum and plasma specimens collected from 6 patient visits over 12 weeks) were derived from 63 individuals who were infected with COVID-19 between February–March 2020. These specimens support the development of COVID-19 antibody tests. Because the second cohort was part of a research study, participant consent was required to allow their samples to be collected and stored in the COVID-19 biorepository. Collected blood samples were packed on ice, couriered directly to the PATH laboratory, and immediately cataloged and processed with universal precautions, even during the citywide lock-down (Fig 1). Some of the modifications to procedures that were implemented due to the lockdowns include, alternative transportation logistics as city buses were disrupted, use of additional personal protective equipment, social distancing procedures to limit the number of people in the lab at one time as well as to space out individuals in the lab, and creating designated work groups to assist in contact tracing if an exposure occurred.

**Logistics.** A logistics plan for managing requests was created including a dedicated email alias (specimenrepository@path.org) to triage and ensure timely responsiveness to inquiries, as well as streamline the process for review and fulfillment of specimens. The intake process started with an intake form linked on the Washington COVID-19 biorepository landing page for interested parties to submit their requests, provide information on their primary research goals, specify the types of specimens requested, and detail the commercialization of technology and tools to be developed. The intake form also noted our BGP's scientific and ethical

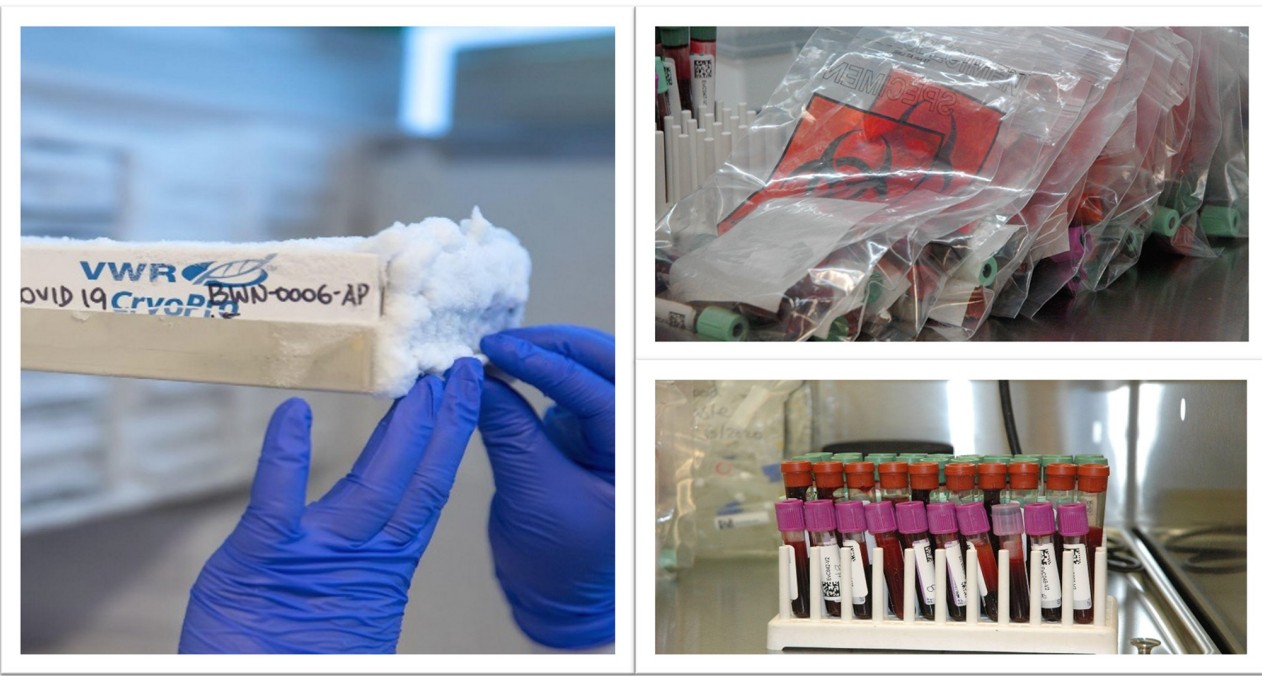

**Fig 1. Biospecimens were immediately received, cataloged, and processed for freezer storage.**

requirements. Completed intake forms were reviewed by the governance committee with technical, ethical, and commercial considerations for approval. Following request approval, and execution of an MTA, the appropriate specimens were pulled from the laboratory freezer and packaged for biological material shipment including proper containment and dry ice. An existing shipper provided overnight shipping in compliance with applicable regulations. Recipients of specimens from the biorepository were responsible for shipping fees.

**Dissemination.** To generate awareness and demand for the biorepository, particularly reaching developers in LMIC settings, a landing webpage was launched to promote the biorepository and its goals. Traffic was driven to the landing page via email campaigns, social media, a press release, and a webinar. Digital marketing efforts included social media, blog posts, and media outreach. Email distributions were used to target cohorts by geography, translating into various languages and further distributed by PATH country offices.

## From biospecimens to RDT images, expanding repositories to include new forms of clinical information

Because of PATH's existing projects and partnerships, access to COVID-19 RDT manufacturers and products is currently being leveraged to advance an RDT image repository [14]. Large diagnostic companies (e.g., Abbott and Quidel) are starting to develop RDT readers and apps to be deployed alongside RDTs for Covid19 diagnostic tools. These combination diagnostic test and digital technology products present unique challenges from a regulatory pathway perspective, especially for smaller manufacturers who may lack the resources and regulatory experience to navigate critical requirements for submissions such as World Health Organization (WHO) Emergency Use Listing (EUL) and Pre-qualification (PQ). Several technical documents are required as part of the PQ submission process including the Design History File,

Device Master Record, quality management system, device specification, and description of manufacturing site, to name a few. Understanding the regulatory landscape better and sharing these processes and learnings to enable more rapid WHO EUL or PQ approval of combination products helps accelerate new test designs.

Additionally, libraries of well-curated, catalogued images of RDTs are critical to enable a range of developers to design, train, and verify machine learning algorithms to correctly score positive and negative samples. Images are being collected and annotated using a range of image capture methods such as Android and iOS devices, under a range of lighting conditions to enable a more robust collection. Contrived (in silico or synthetic) images are also being made to further increase and diversify the repository with varying signal intensity, background, or lighting conditions. The development of the structure of the image repository is drawing on previous biorepository experiences generating, annotating, storing, and granting access to clinical specimens. Working closely with developers to identify how the image libraries should be organized for ideal utilization in app development and verification is important. The repository architecture is also being designed to allow third-party contributions to enable expansion and adaptation as product development evolves. The intent of all of PATH's repositories are to make resources available for free to developers who are qualified, competent, and committed to Global Access provisions.

## Adding video data to repositories advances development of remote sensing of clinical information

Hypoxemia, a common symptom of Covid-19, can be measured using a non-invasive pulse oximeter (PO), and while pulse oximeters have been around for decades, access and awareness of pulse oximetry increased worldwide due to the Covid-19 pandemic [15]. Development of next-generation, or multimodal PO devices was also accelerated, primarily focused on adults. Multimodal PO device manufacturers are leveraging the photoplethysmography (PPG) data used for oxygen saturation measurement to add functionality such as respiratory rate measurement, though other potential parameters include heart rate, anemia and blood pressure [16]. The first multimodal pulse oximeter to measure oxygen saturation and respiratory rate received FDA approval and listing on the United Nations Children's Fund supply catalogue recently, and additional competitors are in the process of entering the market (https://www.masimo.com/products/continuous/rad-g/).

In addition to medical device sensors, there is interest in advancing research on using existing smartphone sensors to measure PPG directly from a video. As with other medical devices, a key barrier that manufacturers have flagged in advancing multimodal POs is the difficulty of validating new devices or algorithms, particularly among children. Respiratory rate is an important clinical measurment in detecting pneumonia, especially in children, but it is not done as often as it should be and is highly variable when manually counted [17–20]. Though children would benefit greatly from better triage and disease management tools, these validation studies are costly, the potential demand for this class of products is unproven, and the global health market is highly price sensitive. As part of an ongoing study among children under five years, PATH is developing an open-access data repository to support the development of integrated primary healthcare clinical measurement tools, particularly for children. Among the data that is being collected across four countries is reference measurements for pulse oximetry, heart rate, respiratory rate, hemoglobin, and PPG, along with deidentified video and audio data files (Fig 2). Additionally, the data repository will follow FAIR Principles (Findability, Accessibility, Interoperability, and Reusability) [21]. The structure will be developed to follow general data modeling rules that reflect best practices for creating effective and

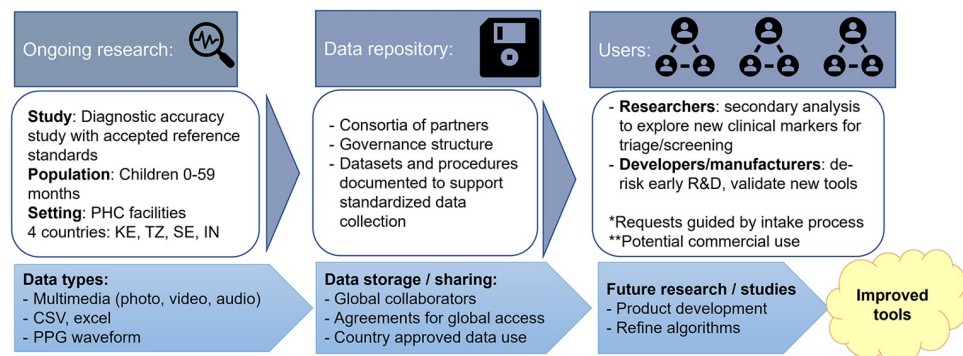

**Fig 2. Example of an open-access data repository structure to support high-quality and responsibly collected data is available for future development of public health products.** There are key considerations as clinical samples or data are transferred from collection to storage to users. This flow diagram maps details for a specific PATH-led study; however, the attributes are relevant to other repository efforts as well. Early planning for the sources, structure, and purpose of the repository helps inform later decisions on execution of the work.

efficient data models. This will include considerations such as identifying the entities and relationships that need to be represented, choosing appropriate data types and constraints, and defining clear and consistent naming conventions so the dataset could be merged with a larger system if required.

## Results and discussion

### Early outcomes of the COVID-19 biorepository

Since March 2020, the Washington COVID-19 biorepository has provided qualified sets of clinical samples to assist in the verification and validation of diagnostic tests. To date, 63 total submissions have been received from 12 different countries, of which 45 were approved, supporting 5 FDA EUA and 1 WHO EUL regulatory pathways. Pre-approval, some reasons for declining requests included a lack of inventory of the requested sample, misalignment between technology stage of development and value/scarcity of sample type, and the requesting organization not having a BSL-2 laboratory, which was a safety requirement. Post-approval, some requests were later "declined" due to an organization's inability to meet legal terms around global access and international shipping restrictions, which in cases prevented import of samples. Improving access to qualified clinical samples allowed developers to better compare tests and identify those best suited for use in a particular environment. This has enabled public health leaders to determine how tests might work together or help troubleshoot challenges as they are deployed in the health system. Some partners who received specimens were able to detect and resolve required modifications for several of their diagnostic tests before deployment in a public health setting, for example ensuring detection of newer variants, and confirming performance of tests across transport media types and cycle thresholds.

### Lessons learned in developing a biorepository for clinical specimens

Despite experience in developing and managing biorepositories, creating one at the start of a pandemic to be utilized as quickly as possible presented challenges. Even with the existing structures and guiding frameworks available as a resource, the team had to work closely with colleagues in facilities, research, and legal departments to address issues quickly and to ensure the safety of staff, the protection of human subjects, adherence to applicable laws, and

appropriate compliance to partner and donor obligations with respect to specimen transfer, use, ownership, and global access requirements. A risk assessment plan was drafted to allow individuals back in the lab in small groups, requiring approval from PATH institutional leadership and further training for team members on updated COVID-19 processing requirements and lab safety before onsite lab access was re-granted.

Careful consideration of human subjects' protection and legal ownership of specimens were also central to this effort, and individually evaluated for each source of samples to ensure alignment with regulatory, legal, and ethical requirements. For samples collected prospectively by other institutions through HSR, consent documentation in those studies had to allow for future use of participant's biological samples for COVID-19 research. Even though PATH was not engaged in the research, ORA provided recommendations aligned with FDA and Common Rule regulations and guidance to ensure samples could be used for product development and commercialization efforts. As a result of this work, PATH also developed a process for utilizing clinical discard samples in future research that falls under FDA guidance.

Legal agreements set the terms for how PATH could receive and share partner samples so developing appropriate legal instruments was an integral step to the functioning framework of the biorepository. Both receiving and sharing samples required an MTA, which needed to have aligned terms, including donor obligations global access terms to ensure the biorepository supported access and availability of high-quality diagnostics to LMICs. Despite the usefulness of a template MTA, many partners did not have legal teams to advise on the documents. To address this challenge, an approach was adopted to make MTAs as flexible and accessible as possible for organizations with limited legal tools. This right-sizing approach also increases inclusiveness by supporting partners to fill gaps in capabilities, without sacrificing quality or safety.

Further challenges of scaling and ensuring a sustainable approach to managing a biorepository were highlighted in this effort, including long-term storage and funding, ensuring visibility to those who can best utilize the materials, and the potential value of global virtual biorepository networks. Significant operational agility, collaboration, and coordination across several stakeholders as well as local partnerships built over years were key to the development of this global good.

## Critical data beyond biospecimens

Even in 2023, the biorepository continues to receive a handful of requests each year. And while new clinical samples have not been added recently, the focus has shifted to electronic data in the form of RDT images. Access to this information enables partners to develop digital readers and machine learning algorithms that support test result interpretation (i.e., controlled images taken under varying conditions can be used to train computer vision models). This work aims to support manufacturers as they incorporate RDT readers into diagnostic products for COVID-19, as well as other diseases like HIV and eventually malaria. Currently, the COVID-19 dataset contains 32,000 images and associated metadata, including a library of 24,000 images that can be used by developers to train machine learning algorithms and around 8,000 images available for algorithm validation purposes. The HIV dataset contains 12,000 images, each with an accompanying annotation file containing 12 attributes. The recently disseminated target product profile for readers of rapid diagnostic tests, led by WHO, speaks to the importance of this companion tool in promoting more consistent, accurate test performance, interpretation, and reporting [22].

A critical issue in diagnostics and AI model development is that the tools developed are only as good as the panels or data used to create them. During the Covid-19 pandemic, a

previously known issue around the accuracy of pulse oximeters across varying skin pigmentation caused renewed concern among healthcare providers, as well as the FDA [23–25]. Technologies that use light reflection and absorption sensors may be affected by skin pigmentation. If manufacturers have not validated their devices among diverse populations of users, then that variability is not well defined and adjusted for. This speaks to the need to ensure diverse user groups are included in device development and validation. Particularly for machine learning, diverse algorithms require diverse datasets to train them. And when those populations are harder to reach, such as children, collaborations are key to ensuring access to high quality and responsibly obtained data.

Another challenge that arises as we transition from freezer storage to cloud storage is how to store and share large data files most efficiently. Larger datasets are needed for machine learning advances, and those datasets will likely include multimodal data sources. A PPG waveform is one example of a data source that includes many data signals per second, and AI will be useful to find meaningful patterns in the signals. Similar to accelerating open access publications, there may be a role for donors to play in accelerating the access and benefits of open access data repositories. For example, offsetting the costs of cloud storage or developing common platforms for use by researchers, could make these global public goods more accessible and quickly.

As newer diagnostic technologies advance, such as connected medical devices or smartphone-based tools, which detect PPG measurement through existing sensors, new ways to accelerate the development, validation, and refinement of important public health products by more manufacturers will ensure a healthy ecosystem for innovation and that the potential benefits of these new tools are able to reach everyone. For example, developer groups are driving cutting edge research in health sensing technologies that use a standard smartphone-based platform. While consumer products are not replacements for medical devices, basic health information that is being derived from these tools is critical for frontline health workers to inform decision-making [26]. For community and primary health care settings, easily accessible and objective health information is valuable for identifying more critically ill or at-risk patients, referring them to necessary care faster, and directing scarce health resources to where they are needed most. Additionally, as smartphone-based sensors become more advanced [27], it is important to consider the potential role of smartphone-based clinical screening tools to digitize health data, enable risk-based stratification of patients, and more rapidly contribute to decision-making on individual and population-based care. With a market penetration of more than 6 billion users, smartphones could be the most readily available health tool already in the pockets of most providers, health workers, and patients globally [28].

## Conclusion

The right diagnostic tools save lives if they are made widely accessible, especially in under-resourced settings across all countries. Through repositories of specimens and data, PATH cost-effectively increases access to resources that are central to the development of promising diagnostic tests for COVID-19 and other health areas. As diagnostic technologies evolve, the need for more diverse and data rich repositories will be key to continue to expand diagnostic product classes and ensure these tools perform as expected everywhere. Strengthening resources to develop appropriate diagnostic tools for high-priority pathogens is critical to more effectively controlling epidemics and pandemics in the future, as well as addressing endemic health conditions. Building partnerships to offer annotated and curated collections for free defrays early R&D risks and costs to support global access and test affordability, ultimately contributing to better health equity.

## Supporting information

**S1 Text. A version of the biorepository governance plan as an example.**
(PDF)

## Author Contributions

**Conceptualization:** Roger Peck, Helen L. Storey, Becky Barney, Shirli Israeli, Olivia Halas, Deborah Oroszlan, Shiri Brodsky, Neha Agarwal, Eileen Murphy, Mariana Sagalovsky, Jessica Cohen, Elizabeth Trias, Aaron Schutzer, David S. Boyle.

**Data curation:** Roger Peck, Helen L. Storey, Becky Barney, Shirli Israeli, Olivia Halas, Deborah Oroszlan, Shiri Brodsky, Neha Agarwal, Eileen Murphy, Mariana Sagalovsky, Jessica Cohen, Elizabeth Trias, Aaron Schutzer, David S. Boyle.

**Funding acquisition:** David S. Boyle.

**Methodology:** Roger Peck, Helen L. Storey, Becky Barney, Shirli Israeli, Olivia Halas, Deborah Oroszlan, Shiri Brodsky, Neha Agarwal, Eileen Murphy, Mariana Sagalovsky, Jessica Cohen, Elizabeth Trias, Aaron Schutzer, David S. Boyle.

**Resources:** Roger Peck, Helen L. Storey, Becky Barney, Shirli Israeli, Olivia Halas, Deborah Oroszlan, Shiri Brodsky, Neha Agarwal, Eileen Murphy, Mariana Sagalovsky, Jessica Cohen, Elizabeth Trias, Aaron Schutzer, David S. Boyle.

**Writing – original draft:** Roger Peck, Helen L. Storey, Becky Barney, Shirli Israeli, Olivia Halas, Deborah Oroszlan, Shiri Brodsky, Neha Agarwal, Eileen Murphy, Mariana Sagalovsky, Jessica Cohen, Elizabeth Trias, Aaron Schutzer, David S. Boyle.

**Writing – review & editing:** Roger Peck, Helen L. Storey, Deborah Oroszlan, David S. Boyle.

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
