## [Decision Letter · Decision Letter 0]

24 Apr 2023

PGPH-D-23-00417

From biorepositories to data repositories: open-access resources accelerate early R&D and validation of equitable diagnostic tools.

Dear Dr. Storey,

Thank you for submitting your manuscript to PLOS Global Public Health. After careful consideration, we feel that it has merit but does not fully meet PLOS Global Public Health’s publication criteria as it currently stands. Therefore, we invite you to submit a revised version of the manuscript that addresses the points raised during the review process.

We look forward to receiving your revised manuscript.

Kind regards,

Dan Kajungu, PhD

Academic Editor

Journal Requirements:

Additional Editor Comments (if provided):

Authors can respond to the comment from reviewers.

Reviewers' comments:

Reviewer's Responses to Questions

**Comments to the Author**

1. Does this manuscript meet PLOS Global Public Health’s publication criteria? Is the manuscript technically sound, and do the data support the conclusions? The manuscript must describe methodologically and ethically rigorous research with conclusions that are appropriately drawn based on the data presented.

Reviewer #1: Partly

Reviewer #2: Yes

2. Has the statistical analysis been performed appropriately and rigorously?

Reviewer #1: N/A

Reviewer #2: N/A

3. Have the authors made all data underlying the findings in their manuscript fully available (please refer to the Data Availability Statement at the start of the manuscript PDF file)?

Reviewer #1: No

Reviewer #2: No

4. Is the manuscript presented in an intelligible fashion and written in standard English?

Reviewer #1: Yes

Reviewer #2: Yes

5. Review Comments to the Author

Reviewer #1: The authors of this manuscript (more a research methods paper, not a research paper per se) present a use case describing the considerations in developing a coordinated open-access specimen biorepository and data registry that can prioritize access and use for innovative research, development and validation of diagnostic tools. Distinguishing these efforts from other ‘academic biorepositories’ is, in part, the desire to support research applications amenable to translation to low and middle income countries, which in turn prompts special considerations in the implementation strategy. A critical point, made in several sections, is the importance of existing partnerships and thereby implicitly sustainability as a Gates Foundation-supported non-profit. More specific details on implementation and operations and on success in achieving its mission goals would be very helpful. The value of a capacity like PATH is an important message for the Global Health community,

STRENGTHS OF THE PAPER; Major Points for improvement

• The paper underscores the importance of pre-existing partners in addressing emergent public health emergencies. Through such relationships, organizations like PATH have developed the internal workflows and governance considerations that offer agility and speed in response.

o The partnerships established by PATH and reflected in this paper are quite diverse (academic health centers, non-profit labs, health departments, local clinics). What was the incentive to send their specimens to PATH? Articulating both the financial and non-financial incentives could help readers approach building similar relationships and infrastructure.

• The authors emphasize the importance of diversity at every step, including specimens and data themselves as well as in the approaches to test and validate technology. Collaborations are key to accessing ‘hard to reach’ and vulnerable populations to make sure that research discoveries pertain to diverse populations. Were these relationships already in place prior to COVID-19?

• The authors identified key considerations in the establishment, implementation and sustainability of the resources (biorepository, data registry) – e.g., IRB, freezer space, MTAs, laboratory capacity (BSL2) –

o More specific insight into the branch point decisions that must be made in developing the structure if a group wished to implement at their site would be helpful. For example, are there different documentation requirements for accepting prospective research specimens vs remnant clinical specimens? is there a ‘go/no go’ determination of whether a new partner is onboarded? Are there ‘checks’ in assessing the rigor and consistency in collecting & handling specimens (the latter could be really important in downstream use and could represent variability in results)? Are there staffing considerations that stem from decisions about strategy and scale? Are there infrastructure considerations? More specificity on the “issues” which can be anticipated and the authors’ team approaches to safety, IRB, legal, MTA, etc. (lines 227-230) would be helpful..

To the latter point about “issues” – which of these were uniquely COVID-19 related (and may substantiate a “pandemic preparedness special consideration” subset) and which are generalizable to repositories/registries built outside of a public health emergency? How will these insights offer PATH and everyone that reads this paper greater agility to navigate the issues more effectively next time?

o The comment that “many partners did not have legal teams” (line 248) was especially meaningful and important as the U.S. research enterprise, broadly writ, is encouraged to expand the reach to underrepresented populations (e.g., via rural hospitals, federally qualified health centers, community locations).

• The authors illustrate the flexible value proposition of their repository by pointing to end users that were able to “detect and resolve required modifications”. More detail would again be helpful as it would set up nicely the premise that “but for the use of PATH biospecimens, these technologic innovations would have been delayed, diminished or deterred”. Detail would elevate the claim from sweeping generalization to evidence-based insight.

• PATH appears to prioritize research and innovation that will make diagnostic tools (and other clinical applications) more accessible to low and middle income countries. The core messages of this manuscript are just as relevant to rural settings in all countries and reinforce important considerations for the research infrastructure everywhere as our ability to conduct experiments at greater scale continues to grow.

WEAKNESSES OF THE PAPER: Additional points for improvement

• Figure 1 provides little useful information and can be deleted.

• The general framework of the first half of the paper is straightforward; however, the pivot (line 185) to applications (pulse oximeters) seems awkward. Can the authors make the transition from biospecimen to physiological measurement devices easier to grasp?

o COVID-19 is caused by the SARS-CoV-2 virus… COVID-19 can lead to hypoxemia. The latter is not the cause of the illness, but is symptomatic of it (line 186)

• Figure 2 needs a legend in order to be clear about the message intended.

o It is also not explained how the approach “ensures high-quality and responsibly collected data” – by what methods is this ensured? What metrics are used to determine quality? How are issues like missing data or interoperability addressed?

• In the context of the data registry description (and data repository of Figure 2), the absence of comment about data models (e.g., OMOP) and FAIR principles that ensure that data from multiple sources can be integrated meaningfully is surprising. This deserves comment.

• Every biorepository of this nature has a limited number of samples The authors mention a request process (63 asks, 45 approved), but there was little explanation of if/how the group prioritized requests to maximize/extend the lifetime of the repository. On what basis were ~1/3 of the requests rejected, if other than lack of mission alignment to address LMIC goals?

o Did mechanical issues like freeze/thaw cycles factor in the dissemination process?

• The specimens were required to be de-identified and remnant specimens (the largest fraction of the biorepository) would have very little accompanying metadata. So, the process for characterization of “qualified, pedigreed” samples is very unclear… did PATH collect information about familial health?

FORMATTING CONSIDERATIONS

• The authors use a lot of acronyms that are not defined (e.g., RDT, LMIC, ORA) which makes reading very difficult. The acronyms should be defined.

• For those unfamiliar with PATH, it reads like another undefined acronym at (its mission is not defined until line 70). The authors should acknowledge that PATH is a non-profit organization committed global health equity issues by fostering innovative and accessible solutions in line 23 and 70.

• The quote in the first sentence of the introduction does not have a ‘close quote’ demarcation, making it unclear how much of that sentence is attributed to the reference.

Reviewer #2: The work shares experiences and lessons learnt relevant for future development efforts of Biorepositories and some immediate outcomes. I commend the authors for the well written article. I have a few comments here below.

1. Ethics statement – Given that the ethical clearance was for the repository being developed, it would be more meaningful to present it as one of the key steps needed for the development of a Biorepository.

2. Lines 135-136- The authors state that “samples were packed, couriered directly to PATH and immediately cataloged and processed with universal precautions even during citywide lockdown”. They however do not state how they dealt with these restrictions to be able to go about their work. -these experiences are important to highlight to inform future work amidst similar pandemics.

2. Although the establishment leveraged on existing PATH resources, it would be important to indicate cost considerations and how these were managed. For example, how much of the total establishment costs, at least an estimate, were covered by existing infrastructure etc.

3. I find that the information on stakeholders and their roles have been presented in very broad terms. The governance plan provides some more overview but again this has its own appendices that are not accessible. I would have wanted to see more details on key stakeholders in the development process beyond collaborations, governance etc. for example who they were and what were their roles in the establishment of the repository.

4. lines 185-208. There is a long narrative under the section “Advancing PPG-derived technology through COVID-19” -that describes this new technology and how relevant it is to COVID19 testing. I however, did not understand how it fits with the “experience for repository development” that can be learnt. I would think that it is diagnostic technology related work not repository development.

5. Finally, I also think that one of the early outcomes of the Biorepository development is the data and metadata being created that can be used by developers to train their machine learning algorithms. This perhaps should have been highlighted as one of the low hanging fruit from this effort.

6. PLOS authors have the option to publish the peer review history of their article (what does this mean?). If published, this will include your full peer review and any attached files.

**Do you want your identity to be public for this peer review?** For information about this choice, including consent withdrawal, please see our Privacy Policy.

Reviewer #1: No

Reviewer #2: No

---

## [Editor Report · Decision Letter 1]

1 Jun 2023

From biorepositories to data repositories: open-access resources accelerate early R&D and validation of equitable diagnostic tools.

PGPH-D-23-00417R1

Dear Helen,

We are pleased to inform you that your manuscript 'From biorepositories to data repositories: open-access resources accelerate early R&D and validation of equitable diagnostic tools.' has been provisionally accepted for publication in PLOS Global Public Health.

Best regards,

Dan Kajungu, PhD

Academic Editor